



# Insights into the sources of ultrafine particle numbers at six European urban sites obtained by investigating COVID–19 lockdowns

Alex Rowell[1], James Brean[1], David C.S. Beddows[1], Zongbo Shi[1], Tuukka Petäjä[2], Máté Vörösmarty[3], Imre Salma[4], Jarkko V. Niemi[5], Hanna E. Manninen[5], Dominik van Pinxteren[6], Roy M. Harrison[1,7], Thomas Tuch[6], Kay Weinhold[6]

[1]School of Geography, Earth & Environmental Sciences, University of Birmingham, Birmingham B15 2TT, United Kingdom
[2]Institute for Atmospheric and Earth System Research (INAR) / Physics, Faculty of Science, University of Helsinki, Finland
[3]Hevesy György Ph. D. School of Chemistry, Eötvös Loránd University, Budapest, Hungary
[4]Institute of Chemistry, Eötvös Loránd University, Budapest, Hungary
[5]Helsinki Region Environmental Services Authority (HSY), Helsinki, Finland
[6]Leibniz Institute for Tropospheric Research (TROPOS), Atmospheric Chemistry Department (ACD), Permoserstr. 15, 04318 Leipzig, Germany
[7]Department of Environmental Sciences, Faculty of Meteorology, Environment and Arid Land Agriculture, King Abdulaziz University, Jeddah 21589, Saudi Arabia

*Correspondence to*: Roy M. Harrison (r.m.harrison@bham.ac.uk)



**ABSTRACT**

Lockdown restrictions in response to the COVID–19 pandemic led to the curtailment of many activities and reduced emissions of primary air pollutants. Here, we applied Positive Matrix Factorization to particle size distribution (PSD) data from six monitoring sites (three urban background and three roadside) between four European cities (Helsinki, Leipzig, Budapest, and London) to evaluate how particle number concentrations (PNCs) and their sources changed during the respective 2020 lockdown periods compared to the reference years 2014–2019. A number of common factors were resolved between sites, including nucleation, road traffic semi–volatile fraction (road traffic$_{svf}$), road traffic solid fraction (road traffic$_{sf}$), diffuse urban (woodsmoke + aged traffic), ozone–associated secondary aerosol (O$_3$–associated SA), and secondary inorganic aerosol (SIA). Nucleation, road traffic, and diffuse urban factors were the largest contributors to mean PNCs during the reference years and respective lockdown periods. However, SIA factors were the largest contributors to particle mass concentrations, irrespective of environment type. Total mean PNCs were lower at two of the urban background and all roadside sites during lockdown. Nucleation and road traffic$_{svf}$ factors response to lockdown restrictions were highly variable, although road traffic$_{sf}$ factors were consistently lower at roadside sites. The responses of diffuse urban factors were largely consistent and were mostly lower at urban background sites. Secondary aerosols (O$_3$-associated SA and SIA) exhibited extensive reductions to their mean PNCs at all sites. These variegated responses to lockdowns across Europe point to a complex network of sources and aerosol sinks contributing to PSDs.



## 1. INTRODUCTION

The COVID–19 pandemic and the resultant curtailment of human activities had profound impacts on global atmospheric chemistry, reflected in the concentrations of greenhouse gases (Le Quéré et al., 2020), gas phase pollutants (Shi et al., 2021), and particulate matter (Hammer et al., 2021; Putaud et al., 2023; Torkmahalleh et al., 2021). Ambient atmospheric aerosol particles are of scientific concern due to their detrimental effects on human health (Cohen et al., 2017) and the uncertainties they cause in models of global radiative forcing (Storelvmo et al., 2016). These health and climatic effects depend on particle size, as their ability to enter the lung, reflect or refract solar radiation, and form clouds, are size dependent processes. For their parameterisation in air quality and climate models, it is therefore important to understand the sources of different sized aerosol particles. Particle mass concentrations (PMCs) are typically dominated by larger particles, such as those with mobility diameters >100 nm, whereas, particle number concentrations (PNCs) are typically dominated by smaller particles, such as those ≤100 nm, commonly referred to as ultrafine particles (UFP). The human health effects of UFP are less clear, however; there is epidemiological evidence for the adverse health effects of UFP exposure when weighted by number (Ohlwein et al., 2019), and particle count in different size ranges is an important metric for understanding the climatic effects of aerosols (Jiang et al., 2021).

Number size distributions are typically comprised of a series of lognormal modes. Each mode represents a different source, or aggregate of sources, (referred to as a factor) as modified by aerosol microphysical processes, such as particle shrinkage. Different sources of particles produce different modal diameters (Vu et al., 2015). In urban environments, primary and secondary particle production (<1,000 nm) arises from factors of natural and anthropogenic origin. In the existing literature, a number of common source–related factors have been identified around the world, including nucleation, traffic (multiple), heating, ozone–associated secondary aerosol, and secondary inorganic aerosol, as well as biomass burning, and various unidentified factors (Hopke et al., 2022). However, the identification of such factors is not straightforward. The shape of the particle size distribution from primary traffic emissions, for example, depends on a host of variables, including fuel type, and driving conditions (Rönkkö and Timonen, 2019). Secondary aerosols arise in the smallest diameters from new particle formation (NPF) processes, where vapours such as sulphuric acid and amines cluster in the atmosphere to form new thermodynamically stable aerosol particles at ~1.5 nm, before growing to larger sizes due to the condensation of oxidised organic molecules (OOMs), acids, and other suitably involatile vapours (Kulmala et al., 2014; Lee et al., 2019). Vapour nucleation and the subsequent formation of new particles occurs most commonly during photochemically active periods. Secondary accumulation mode aerosols are typically comprised largely of nitrate, sulphate, organic matter, and





ammonium, and the modal diameter depends on precursor concentrations and rates of oxidation chemistry. Whilst size distribution data does not provide any compositional information, sources and mechanisms can be inferred through their modal diameters, as well as their daily, weekly, and monthly cycles.


The COVID–19 pandemic, and the associated lockdown periods, will have impacted these factors and their respective contributions to total PNCs. In order to better understand these changes, a mathematical receptor model (Positive Matrix Factorisation, PMF) was applied to particle size distribution data from six monitoring sites between four European cities (Helsinki, Leipzig, Budapest, and Lon-

don) to identify the major factors contributing to total PNCs during lockdown compared to the associated periods in the reference years 2014–2019, depending on data coverage.

## 2.    METHODOLOGY

### 2.1.    Monitoring Stations

This study is based on data from six monitoring sites spanning four European countries including

Finland, Germany, Hungary, and the United Kingdom (**Figure 1**). Each site was assigned a unique identifier denoting its location and environment type (**Table 1**). The monitoring sites incorporate two different environment types, including urban background (denoted as UB) and roadside (denoted as RS) locations. Three urban background and three roadside sites were included in this study. The urban background sites are not dominated by any single source or street and are deemed representative of a

well–mixed, average atmospheric environment within their respective cities. Their surroundings are similarly varied and broadly consist of commercial and residential property, transport infrastructure, and greenspace. The roadside sites are located such that their pollution levels are heavily influenced by the emissions from nearby traffic. Their surroundings are also similarly varied; however, they differ in terms of their aspect ratios (average building height divided by the most frequent width of

the street canyon) and daily traffic volumes.





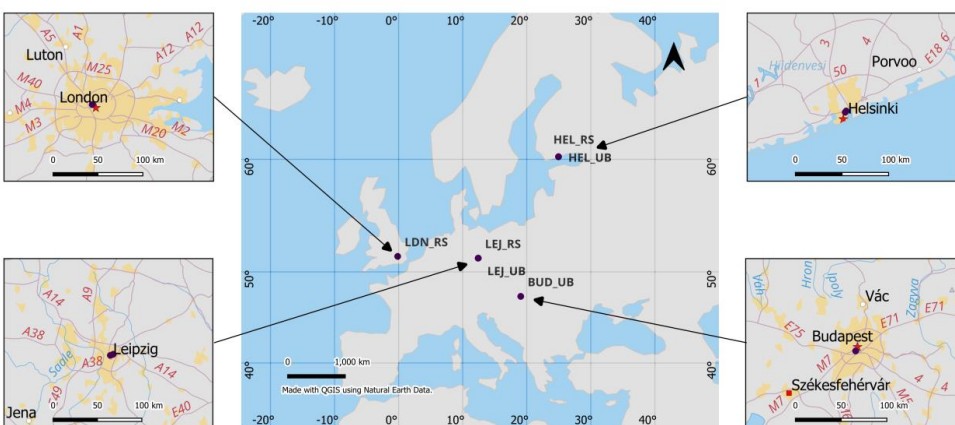

**Figure 1:** Monitoring site locations. *Above left*, London (United Kingdom) roadside; *above right*, Helsinki (Finland) urban background & roadside; *below left*, Leipzig (Germany) urban background & roadside; *below right*, Budapest (Hungary) urban background.


**Table 1:** Monitoring site information. An Asterix (*) in the Site ID column denotes if a site is participating in ACTRiS.

| City (country) | Monitoring site | Site ID | Environment Type | Coordinates and height above sea level | Lockdown periods | Peak traffic hours (local times) |
|---|---|---|---|---|---|---|
| **Helsinki (Finland)** | Kumpula (SMEAR III) | HEL_UB* | Urban background | N60°12'10" E24°57'40" 29 m | March 17th – May 15th | A.M. 08:00 P.M. 16:00 |
| **Leipzig (Germany)** | TROPOS rooftop | LEJ_UB* | Urban background | N51°21'09" E12°26'04" 127 m | March 23rd – May 5th | A.M. 07:00– 08:00 P.M. 15:00–16:00 |
| **Budapest (Hungary)** | Lágymányos (BpART) | BUD_UB | Urban background | N47°28'30" E19°3'45" 115 m | March 17th – May 19th | A.M. 07:00– 09:00 P.M. 15:00–18:00 |
| **Helsinki (Finland)** | Mäkelänkatu | HEL_RS | Roadside | N60°11'48" E24°57'06" 33 m | March 17th – May 15th | As previously stated. |
| **Leipzig (Germany)** | Eisenbahnstrasse | LEJ_RS* | Roadside | N51°20'44" E12°24'23" 120 m | March 23rd – May 5th | As previously stated. |
| **London (United Kingdom)** | Marylebone Road | LDN_RS | Roadside | N51°31'21" W000°09'17" 35 m | March 27th – May 16th | A.M.–P.M. 07:00–20:00 |

Peak traffic hours were obtained from TomTom traffic index **https://www.tomtom.com/traffic-index/**



### 2.1.1. Helsinki, Finland

Helsinki is located in Southern Finland, at the shore of the Gulf of Finland in the Baltic Sea. The
capital is Finland's largest city, with 0.66 million inhabitants or 1.59 million inhabitants inclusive of
its sub–regional units. Finland has 497 passenger cars per 1,000 inhabitants, which is below the av-
erage motorisation rate in the European Union (EU). However, Helsinki's car density is ~1,035 cars
per km$^2$ which is comparable with densities observed in other cities across Europe.

Urban background aerosol data was obtained from the SMEAR III research station (Jarvi et al., 2009),
located in the Kumpula campus of the University of Helsinki. Measurements are taken at 4 m above
ground level and at distances >125 m from highly–trafficked roads bordering the site. The busy road-
ways experience 44,000 vehicles per workday (Järvi et al., 2012). The stations immediate surround-
ings also include multi–storey buildings, access roads, allotted parking bays, and greenspace.


Roadside aerosol data was obtained from a supersite monitoring station along one of Helsinki's main
thoroughfares known as Mäkelänkatu. The street is ~42 m in width and is flanked by three– and four–
storey buildings, yielding an aspect ratio of 0.40 (often referred to as an avenue canyon) (Rönkkö et
al., 2017) and experiences 28,000 vehicles per day (Helin et al., 2018; Kuula et al., 2020). The stations
immediate surroundings also include six lanes of traffic (three in each direction of travel), a central
tramline bordered by tall vegetation, two footpaths, and on–street parking.

### 2.1.2. Leipzig, Germany

Leipzig is located in the German State of Saxony in east Germany. Leipzig is the 8[th] most populated
city in Germany, with 0.6 million inhabitants. Germany has 574 passenger cars per 1,000 inhabitants
which is comparable with the average motorisation rate in the EU. However, Leipzig's vehicle density
is ~913 vehicles per km$^2$ which is comparable with densities observed in other cities across Europe.

Urban background aerosol data was obtained from an atmospheric research station operated by the
Leibniz Institute for Tropospheric Research (TROPOS) within the Leipzig Science Park. Measure-
ments are taken on the roof of an institute building at 16 m above ground level and at distances >100
m from highly–trafficked roads bordering the site (Klose et al., 2009; Birmili et al., 2016). The Sci-
ence Park contains other research institutes and related companies, greenspace, and allotted parking
bays, including a multi–storey carpark. The Park perimeter includes transport infrastructure (road,
rail and tramways), commercial property (restaurants, hotels, a petrol station etc.), residential prop-
erty, on–street parking, and greenspace.





Roadside aerosol data was obtained from a permanent observation site located along an important connecting road in the east of the city known as Eisenbahnstrasse. Measurements are taken from an apartment window at 6 m above ground level on the northern side of the street. The street is ~20 m

in width and is flanked by multi–storey period buildings, yielding an aspect ratio of 0.90, and experiences 12,000 vehicles per working day (Birmili et al., 2016). The stations immediate surroundings also include two–lanes of traffic (one in each direction of travel), an integrated tramline, on–street parking, two bicycle lanes (one in each direction of travel), two footpaths, and scant vegetation.

### 2.1.3.  Budapest, Hungary

Budapest is located in the Carpathian Basin in central Hungary. It is the capital and the largest city of the country, with 1.72 million inhabitants. Hungary has 395 passenger cars per 1,000 inhabitants which is one of the lowest motorisation rates in the EU. However, Budapest's car density is ~1,315 cars per km$^2$ which is comparable with densities commonly observed in other European cities.

Urban background aerosol data was obtained from the Budapest platform for Aerosol Research and Training (BpART) Laboratory, located on the second–floor balcony of the Northern block in the Lágymányos Campus of the Eötvös Loránd University. The balcony is 11 m above the street level of the closest road and is situated 85 m from the right bank of the River Danube (Salma et al., 2016). Sampling inlets and sensors are set up at heights of between 80 and 150 cm above the rooftop level

of the measurement container.

### 2.1.4.  London, United Kingdom

London is located in southeast England (United Kingdom, UK). The capital is the UKs largest city, with 9 million people occupying greater London. The UK has 589 passenger cars per 1,000 inhabitants which is comparable with the average motorisation rate in the EU. However, Greater London's

car density is ~1,938 cars per km$^2$ which is well above average densities commonly observed in other cities across Europe.

Roadside aerosol data was obtained from a supersite monitoring station, located along Marylebone Road opposite one of London's top attractions, Madame Tussauds. Measurements are taken on the

roof of a cabin positioned kerbside at 4 m above ground level (Harrison et al., 2019). The street is ~34 m in width and is flanked by multi–storey buildings, yielding an aspect ratio of 1.00 (often referred to as a regular street canyon), and experiences 80,000 vehicles per day (Harrison et al., 2019). The monitoring stations surroundings also include six lanes of traffic (three in each direction of





travel), two footpaths, and scant vegetation. Analyses of air quality data from this site have been

reported by Kamara and Harrison (2021).

## 2.2. Instrumentation and Data Coverage

Instrumentation used to sample aerosol size distributions, as well as gaseous pollutants, particle mass
concentrations, black carbon concentrations, and vehicle counts (referred to as auxiliary data) at the
different monitoring sites are stated in **Table 2**, their sampling methodologies are outlined in **SI sec-**

**tion 1.1**, and the applicable data coverage is plotted in **Figure S1**. The particle sizers covered different
size ranges therefore a reduced common range (10–600 nm) was selected (although the whole range
was utilised in the model; see **SI section 1.2**). The years of study varied depending on the availability
of the data but each dataset covered the associated spring 2020 lockdown period, as well as an equiv-
alent time period between 2014 and 2019, for comparison purposes.


**Table 2:** Sampling equipment used at each monitoring site.

| Site ID | Particle sizer (size range, nm) | Black carbon (BC) | Nitrogen oxides (NO$_x$) | Ozone (O$_3$) | Sulphur dioxide (SO$_2$) | Particulate matter (PM$_{2.5}$) | Vehicle count (VC) |
|---|---|---|---|---|---|---|---|
| **HEL_UB** | DMPS (3–800) | – | Thermo TEI42 | Thermo TEI49 | Horiba APSA 360 | – | – |
| **LEJ_UB** | TDMPS (3–800) | – | Horiba APNA 370 | Horiba APOA–350E | Thermo Electron TE 43C–TL | – | – |
| **BUD_UB** | DMPS (6–1,000) | – | Thermo 42C | Ysselbach 43C | – | Environment MP101M | Traffic counter |
| **HEL_RS** | DMPS (6–800) | MAAP | Horiba APNA–370 | Thermo Electron Model 49i/Horiba APOA–370 | – | Thermo TEOM 1405 | Traffic counter |
| **LEJ_RS** | DMPS / TDMPS (5/10–800) | – | – | – | – | – | – |
| **LDN_RS** | SMPS (16.6–604) | Magee AE33 | Teledyne API 200E | Teledyne API 400E | – | Palas Fidas 200 | – |

## 2.3. Positive Matrix Factorisation

The application of PMF is similar to the previous work of Rivas et al. (2020) and implemented using
PMF2 (Positive Matrix Factorization (PMF2) (Sep 25, 2012) vers. 4.2, Copyright 1993, 2004 Pentti

Paatero, Helsinki, Finland). PMF is a well–established receptor model (Paatero and Tapper, 1994)
used to solve functional mixing models when the source profiles are unknown and presumed to be
constant. PMF solutions are constrained to be non–negative, and a least squares algorithm is applied
which accounts for uncertainties in the dataset. PMF is therefore quantitative and identifies a user–



specified number of sources depending on how well the outputs describe the monitoring site. These make it suitable for the source apportionment of size distribution data. Details of the PMF methodology can be found in **SI section 1.2**.

### 2.4. Condensation Sink

The condensation sink (CS) represents the rate at which a vapour phase molecule will collide with pre–existing particle surface, and was calculated from the size distribution data as follows:

$$CS = 2\pi D \sum_{d_p} \beta_{m,d_p} d_p N_{d_p} \tag{1}$$

where $D$ (m$^2$ s$^{-1}$) is the diffusion coefficient of the diffusing vapour (assumed sulphuric acid), $\beta_m$ is a transition regime correction, $d_p$ (m) is particle diameter, and $N_{dp}$ (m$^{-3}$) is the number of particles at diameter $d_p$.

### 2.5. Statistical Analysis

An Independent Samples T–test was performed to compare the means of the resolved factors from the PMF model (from the source apportionment of size distribution data; **section 2.3**) for the respective lockdown periods at each site to the equivalent periods in the reference years (depending on data availability). The purpose of this test was to determine if the resolved factors were different between the associated periods at each location. The significance level was set at 0.05 (i.e. a p value <0.05 was deemed statistically significant).

### 3. RESULTS AND DISCUSSION

### 3.1. Identification of Major Factors

Data covering the spring 2020 lockdown period for each monitoring site were individually analysed using PMF. Multiple outputs were compared and a solution was selected based on the cogency and spatiotemporal behaviour of each factor. A number of common factors were resolved between the sites including nucleation, road traffic semi–volatile fraction (road traffic$_{svf}$), road traffic solid fraction (road traffic$_{sf}$), diffuse urban (woodsmoke + aged traffic), ozone–associated secondary aerosol (O$_3$–associated SA), and secondary inorganic aerosol (SIA). Despite their commonalities, factors exhibited varying profiles at each site. Their particle number (**Figure 2a**) and mass (**Figure 2b**) size distributions, mean daily (**Figure 3a**) and weekly (**Figure 3b**) cycles, relative associations with available auxiliary data (**Figure 4**), and polar plots showing wind directions and speeds coincidental with the top 25$^{th}$ percentile of factor intensity (**Figure S5**) were tabled and/or plotted for analysis. The factors are broadly summarised below and site–by–site descriptions can be found in **SI section 2.1**.





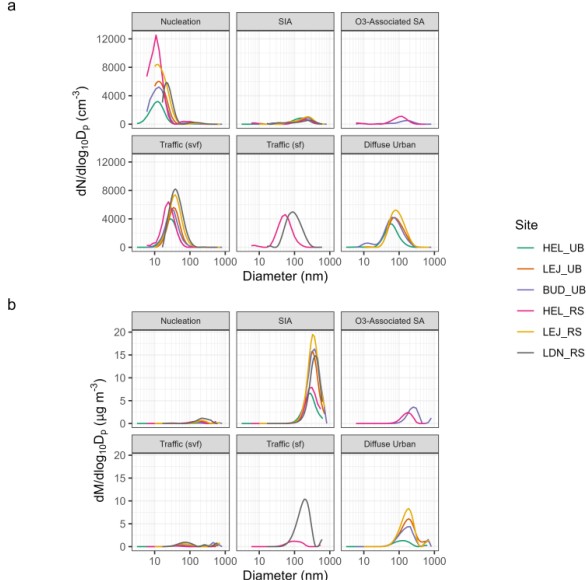

**Figure 2:** Number (a) and mass (b) size distribution data for each factor at each monitoring site. Each panel represents a factor and each colour represents a site.

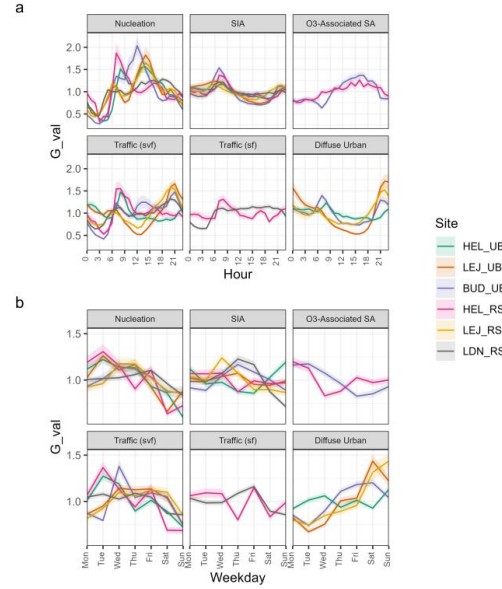

**Figure 3:** Mean G values from the PMF model for each factor, showing (a) daily and (b) weekly cycles. Each panel represents a factor and each colour represents a monitoring site. The shaded region represents the standard error of the mean.





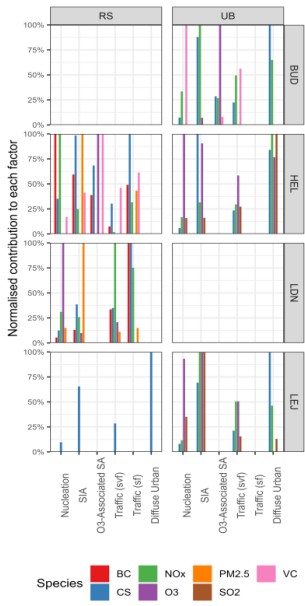

**Figure 4:** Normalised contribution of downweighted auxiliary variables to each factor for each monitoring site. The mean G values for each variable for each site are normalised to a maximum of 1 and expressed as a percentage.

### 3.1.1. Nucleation

Factors attributed to nucleation were resolved at all sites. The factors had a major mode in the size distribution which peaked at ~11–22 nm (**Figure 2a**) and a mass distribution dominated by particles in the accumulation mode (**Figure 2b**). Factor contributions typically peaked in the afternoon and with busy traffic periods (**Figure 3a**), and were higher on weekdays compared to weekends (**Figure 3b**). The factors were associated to varying degrees with BC, $NO_x$, $SO_2$, $O_3$, and $PM_{2.5}$ (**Figure 4**). Additionally, the factors were consistently associated with low CS (**Figure 4**) which is a key determinant for NPF in urban environments (Deng et al., 2021). CS is the largest sink for vapours such as sulphuric acid, as well as other low volatility molecules, and occurs synchronously with high coagulation losses of small particles. We inferred that this factor is likely the sum of particles produced by means of photochemically induced nucleation processes, as well as some combustion–related processes, such as ultrafine vehicle emissions and the formation of new particles through the dilution and cooling of vehicle exhaust (Charron and Harrison, 2003; Pérez et al., 2010). The diurnal profiles at most sites suggest a predominantly non–traffic formation source. However, the nucleation factor at HEL_RS is mainly related to the traffic source, since the diurnal variation shows clear morning and afternoon rush hour peaks. and the contribution of $NO_x$ and BC are very high.



### 3.1.2. Road traffic semi–volatile fraction (road traffic$_{svf}$)

Factors attributed to road traffic$_{svf}$ were resolved at all sites. The factors had a major mode in the size distribution which peaked at ~25–40 nm (**Figure 2a**) and a mass distribution dominated by Aitken and/or accumulation mode particles (**Figure 2b**). Factor contributions typically peaked in the morn-

ing and afternoon/evening (**Figure 3a**), were higher on weekdays compared to weekends (**Figure 3b**), and were associated to varying degrees with combustion–related pollutants including BC, NO$_x$, SO$_2$, and PM$_{2.5}$ (**Figure 4**). In Europe, diesel vehicles are responsible for much of the exhaust PM from road traffic (Damayanti et al., 2023). Particles emitted in diesel exhaust fall into two main categories: semi–volatile and solid graphitic (black carbon) particles (Harrison et al., 2018; Kittelson et

al., 2006). We inferred that these factors likely represent the former particle type, as well as a likely contribution from other mobile and/or stationary combustion–related activities (i.e. cooking and heating emissions). The factors closely resemble those referred to as 'traffic 1' in the literature, typically in reference to spark-ignition vehicle emissions or freshly emitted traffic particles (Hopke et al., 2022) and may include gasoline vehicle emissions.

### 3.1.3. Road traffic solid fraction (road traffic$_{sf}$)

Factors attributed to road traffic$_{sf}$ were resolved at HEL_RS and LDN_RS. The factors had a major mode in the size distribution which peaked at ~55–90 nm (**Figure 2a**) and a mass distribution dominated by particles in the Aitken and/or accumulation mode (**Figure 2b**). Factor contributions typically peaked in the morning and afternoon/evening (**Figure 3a**), were higher on weekdays compared to

weekends (**Figure 3b**), and were associated to varying degrees with combustion–related pollutants including BC, NO$_x$, SO$_2$, and PM$_{2.5}$ (**Figure 4**). We inferred that these factors predominantly represent the solid particle mode arising from diesel road traffic (Harrison et al., 2018; Kittelson et al., 2006). The factors closely resemble those referred to as 'traffic 2' in the literature, typically in reference to diesel vehicle emissions or distant traffic particles (Hopke et al., 2022).

### 3.1.4. Diffuse urban

Factors attributed to diffuse urban were mainly resolved at urban background sites. The factors had a major mode in the size distribution which peaked at ~75–90 nm (**Figure 2a**) and a mass distribution dominated by accumulation mode particles (**Figure 2b**). Factor contributions typically peaked in the morning and evening (**Figure 3a**), were higher on weekends compared to weekdays (**Figure 3b**), and

were associated to varying degrees with combustion–related pollutants, including NO$_x$ and SO$_2$ (**Figure 4**). The factors closely resemble those referred to as 'urban background' by Beddows et al. (2015), and later 'diffuse urban' by Beddows and Harrison (2019), representing aged woodsmoke and road traffic emissions. 'Urban background' has other connotations and therefore we opted to use 'diffuse





urban' as a factor name as it better categorised the emission source, with less confusion amongst the
literature.

### 3.1.5.  Ozone–associated secondary aerosol (O$_3$–associated SA)

Factors attributed to O$_3$–associated SA were resolved at BUD_UB and HEL_RS. The factors had a
major mode in the size distribution which peaked at ~125–175 nm (**Figure 2a**) and a mass distribution
dominated by particles in the accumulation mode (**Figure 2b**). Factor contributions peaked in the
daytime (**Figure 3a**), were higher on weekdays compared to weekends (**Figure 3b**), and were
strongly associated with O$_3$ (**Figure 4**). The exact nature of these factors are uncertain; however, they
are consistent with other such observations (often referred to as O$_3$–rich SA) in the available literature
(Ogulei et al., 2007; Liu et al., 2014; Squizzato et al., 2019) and likely represent particles which have
grown through the condensation of secondary material (Hopke et al., 2022).

### 3.1.6.  Secondary inorganic aerosol (SIA)

Factors attributed to SIA were resolved at all sites. The factors had a major mode in the size distribu-
tion which peaked at ~175–265 nm (**Figure 2a**) and a mass distribution dominated by accumulation
mode particles (**Figure 2b**). Factor contributions typically peaked in the morning and evening/night
(**Figure 3a**), were higher on weekdays compared to weekends (**Figure 3b**), and were associated with
a host of auxiliary variables (**Figure 4**). Multimodal number size distributions have also been ob-
served at other locations and suggest the presence of both local and distant particles thought to have
been formed through the atmospheric processing of NO$_x$ and other gaseous precursor emissions
(Ogulei et al., 2007; Kasumba et al., 2009).

### 3.2.  Changes in Factor Contributions Under Lockdown Restrictions

To evaluate the effects of lockdown restrictions on PNCs, the applicable 2020 lockdown periods at
each monitoring site were compared to the equivalent days of the year in the reference years 2014–
2019, depending on data coverage. In examining temporal changes affecting pollutant concentrations,
it is now common practice to remove the influences of changes in weather variables which affect
primary pollutants concentrations (Vu et al., 2019).  Such a treatment was not applied in this study as
weather variables affect both photochemical nucleation (Bousiotis et al., 2021) and the traffic semi-
volatile fraction (Charron et al., 2003) in complex ways which differ from the influence upon primary
pollutants and hence might produce unpredictable artefacts if applied to particle number distribution
data. Rather, the trends in meteorological variables are presented in **Figure S7**, and commented upon
in the text, where considered relevant.




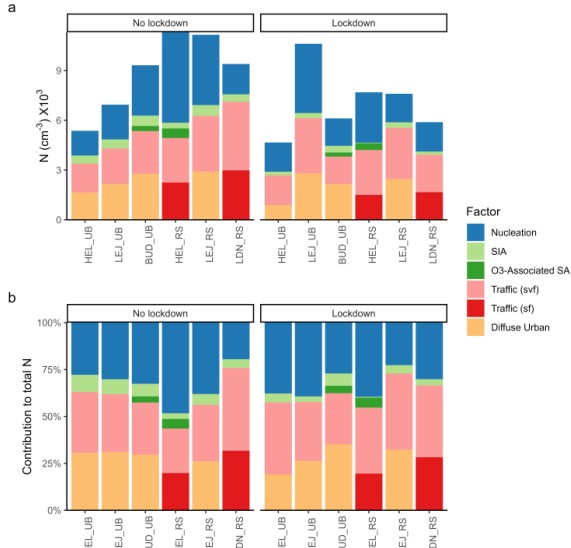

**Figure 5:** Mean total particle number concentrations (PNCs) (a) and mean contributions to total PNCs (b) for the 2020 lockdown period for each factor at each monitoring site and the equivalent periods in the reference years 2014–2019, depending on data availability. Also see Tables 3 and 4.

**Table 3:** Mean and min/max particle number concentrations ($cm^{-3}$) during lockdown compared to the
equivalent periods in the reference years 2014–2019 (depending on data availability) for urban background sites. An Asterix (*) in the Lockdown column denotes if a factor is significantly different (p value is less than 0.05) between the reference years and lockdown period by way of an Independent Samples T–test.

| Factors | HEL_UB | | | | LEJ_UB | | | | BUD_UB | | | |
|---|---|---|---|---|---|---|---|---|---|---|---|---|
| | Reference | | Lockdown | | Reference | | Lockdown | | Reference | | Lockdown | |
| | Mean | Min/Max | Mean | Min/Max | Mean | Min/Max | Mean | Min/Max | Mean | Min/Max | Mean | Min/Max |
| Nucleation | 1,490 | 0/22,495 | 1,754* | 0/14,053 | 2,178 | 0/50,641 | 4,176* | 0/38,805 | 3,037 | 0/42,887 | 1,656* | 0/26,723 |
| Road traffic$_{svf}$ | 1,746 | 0/19,577 | 1,789 | 79/9,659 | 2,211 | 0/19,586 | 3,334* | 0/14,803 | 2,589 | 0/30,302 | 1,665* | 0/14,886 |
| Road traffic$_{sf}$ | – | – | – | – | – | – | – | – | – | – | – | – |
| Diffuse urban | 1,642 | 0/10,943 | 889* | 0/4,969 | 2,172 | 0/36,819 | 2,789* | 0/33,532 | 2,762 | 0/13,977 | 2,150* | 0/8,891 |
| O₃–associated SA | – | – | – | – | – | – | – | – | 307 | 0/1,560 | 245* | 0/1,155 |
| SIA | 492 | 0/3,893 | 221* | 14/1,192 | 500 | 0/2,216 | 318* | 5/1,316 | 626 | 0/2,452 | 402* | 0/1,870 |
| Total | 5,370 | – | 4,653 | – | 6,944 | – | 10,617 | – | 9,321 | – | 6,118 | – |

**Table 4:** Mean and min/max particle number concentrations ($cm^{-3}$) during lockdown compared to the equivalent periods in the reference years 2014–2019 (depending on data availability) for roadside sites. An Asterix (*) in the Lockdown column denotes if a factor is significantly different (p value is less than 0.05) between the reference years and lockdown period by way of an Independent Samples T–test.

| Factors | HEL_RS | | | | LEJ_RS | | | | LDN_RS | | | |
|---|---|---|---|---|---|---|---|---|---|---|---|---|
| | Reference | | Lockdown | | Reference | | Lockdown | | Reference | | Lockdown | |
| | Mean | Min/Max | Mean | Min/Max | Mean | Min/Max | Mean | Min/Max | Mean | Min/Max | Mean | Min/Max |
| Nucleation | 5,464 | 0/65,412 | 3,041* | 0/23,259 | 4,243 | 0/45,090 | 1,721* | 0/19,449 | 1,830 | 0/11,119 | 1,775 | 0/8,393 |
| Road traffic$_{svf}$ | 2,684 | 0/26,689 | 2,704 | 391/12,156 | 3,362 | 0/24,168 | 3,083* | 8/12,747 | 4,145 | 43/16,990 | 2,257* | 0/7,807 |
| Road traffic$_{sf}$ | 2,247 | 0/16,009 | 1,503* | 370/3,718 | – | – | – | – | 2,981 | 0/12,525 | 1,664* | 0/8,318 |
| Diffuse urban | – | – | – | – | 2,907 | 0/38,104 | 2,456* | 88/24,864 | – | – | – | – |
| O₃–associated SA | 579 | 0/2,758 | 409* | 0/1,349 | – | – | – | – | – | – | – | – |
| SIA | 342 | 16/2,702 | 34* | 0/194 | 648 | 0/2,572 | 340* | 11/1,411 | 444 | 16/2,422 | 194* | 0/1,973 |
| Total | 11,320 | – | 7,692 | – | 11,140 | – | 7,600 | – | 9,400 | – | 5,890 | – |




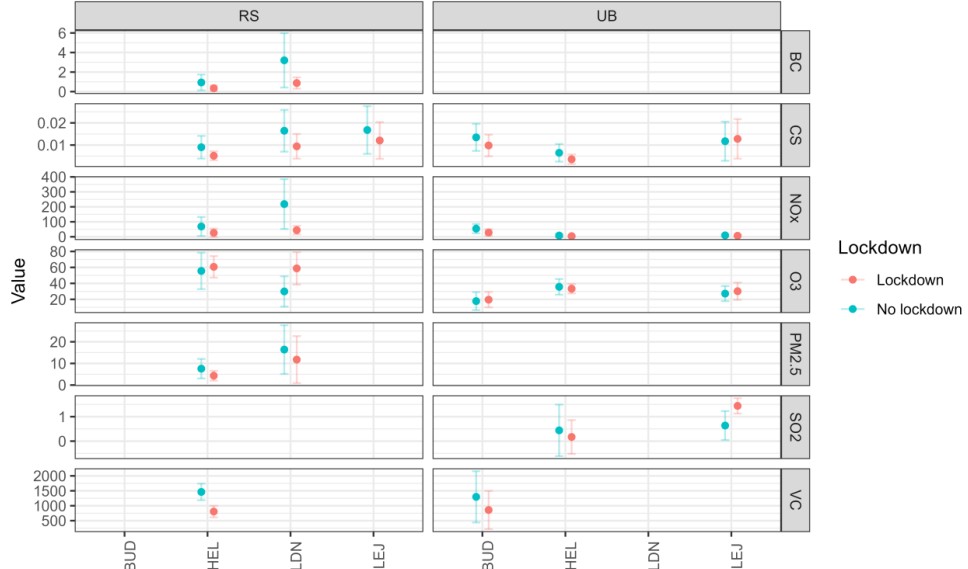

**Figure 6:** Means of auxiliary variables during lockdown compared to the equivalent periods in the reference years 2014–2019 (depending on data availability) for each monitoring site included in this study. Black carbon (BC) = $\mu$g m$^{-3}$, condensation sink (CS) = s$^{-1}$, nitrogen oxides (NO$_x$), ozone (O$_3$), and sulphur dioxide (SO$_2$) = ppb, and vehicle count (VC) = h$^{-1}$.

### 3.2.1. Comparison of total particle number concentrations during lockdowns to the reference years

During lockdown, total mean PNCs were lower at most urban background (**Figure 5** and **Table 3**) and all roadside sites (**Figure 5** and **Table 4**) included in this study, in comparison to the reference years. Regression estimates also showed that measured PNCs were lower than predicted at the majority of monitoring sites (**Figure S6a**).

Among urban background sites, total mean PNCs were higher at LEJ_UB, and lower at HEL_UB and BUD_UB, during lockdown compared to the equivalent periods in the reference years (**Figure 5** and **Table 3**). Nucleation, road traffic$_{svf}$, and diffuse urban factors were primarily responsible for the changes in total mean PNCs at LEJ_UB.

Mean PNCs were lower at all of the roadside sites during lockdown compared to the equivalent periods in the reference years (**Figure 5** and **Table 4**). Changes to nucleation and road traffic factors were primarily responsible for the fall in total mean PNCs at roadside sites.



### 3.2.2. Comparison of road traffic–related particle number concentrations during lockdowns to the reference years

Mean PNCs from factors attributed to road traffic$_{svf}$, during reference and lockdown periods, were amongst the largest of the resolved factors, irrespective of environment type (**Figure 5** and **Table 3/Table 4**). The response of these factors to lockdown restrictions were varied. Mean PNCs from factors attributed to road traffic$_{sf}$ factors were also substantial, regardless of lockdown measures (**Figure 5** and **Table 4**). However, their response to lockdown restrictions were consistent.

Among urban background sites, mean PNCs from road traffic$_{svf}$ factors were higher at LEJ_UB, comparable at HEL_UB, and lower at BUD_UB during lockdown compared to the equivalent periods in the reference years (**Figure 5** and **Table 3**). Large reductions in traffic volumes were observed on major roads across Finland (Riuttanen et al., 2021) and the German state of Saxony (Jaekel and Muley, 2022). However, the associated emissions reductions may have been offset by increased emissions from private households. Fuel oils are regularly used in urban households for a variety of domestic activities and predominantly generate Aitken and nucleation mode particles (Tiwari et al., 2014). Lockdown restrictions had immediate and varied impacts on energy use, with increased residential demand due to people being confined to their homes. Road traffic volumes in central Budapest were also substantially reduced (Salma et al., 2020). However, these reductions were perhaps better reflected in the traffic$_{svf}$ factor at BUD_UB.

At roadside sites, mean PNCs from road traffic$_{svf}$ factors were comparable at HEL_RS, and lower at LEJ_RS and LDN_RS during lockdown compared to the equivalent periods in the reference years (**Figure 5** and **Table 4**). Mean PNCs from road traffic$_{sf}$ factors were also lower at HEL_RS and LDN_RS during lockdown (**Figure 5** and **Table 4**). Changes to mean PNCs from road traffic factors were notably more pronounced at LDN_RS than at other roadside sites included in this study. This is in some measure due to the enormous volumes of traffic typically present along Marylebone Road which were significantly reduced during lockdown (Hicks et al., 2021). However, BC concentrations have reduced considerably in recent years (**Figure S6c**), likely associated with the increased proportion of Euro 6/VI–compliant vehicles (Damayanti et al., 2023; Luoma et al., 2021). The adoption of Euro 6/VI vehicle emissions standards and the compulsory emission technologies (such as diesel particle filters, DPFs) will have impacted road traffic factors over time, especially the road traffic$_{sf}$ factors resolved at HEL_RS and LDN_RS. DPFs do not suppress the semi–volatile mode of the emissions with high efficiency (Damayanti et al., 2023).



### 3.2.3. Comparison of diffuse urban particle number concentrations during lockdowns to the reference years

Mean PNCs from factors attributed to diffuse urban, during reference and lockdown periods, were amongst the largest of the resolved factors (**Figure 5** and **Table 3/Table 4**). The response of these factors to lockdown measures were largely consistent.

Among urban background sites, mean PNCs from diffuse urban factors were higher at LEJ_UB, and lower at HEL_UB and BUD_UB, during lockdown compared to the equivalent periods in the reference years (**Figure 5** and **Table 3**). Mean PNCs from the factor were also lower at LEJ_RS during lockdown (**Figure 5** and **Table 4**). Changes to mean PNCs from diffuse urban factors likely reflect variations in residential wood combustion and reductions to traffic volumes. As previously mentioned, large reductions to traffic volumes were reported across Helsinki (Riuttanen et al., 2021), the German state of Saxony (Jaekel and Muley, 2022), and Central Budapest (Salma et al., 2020). Variations in residential wood combustion, on the other hand, may reflect responses to outdoor temperature fluctuations (**Figure S7**), as well as the more social and cultural aspects of urban air pollution. It is commonplace in Northern countries to use firewood in sauna stoves and various fireplaces as supplementary heating (Kukkonen et al., 2020). Reductions in BC concentrations in Northern Helsinki may be related to decreased wood–burning (Harni et al., 2023) and/or weather conditions during lockdown. Nevertheless, biomass burning has previously been identified as a major contributor to PM concentrations in Helsinki (PM$_{2.5}$; Saarnio et al., 2012), Leipzig (PM$_{10}$; van Pinxteren et al., 2016), and Budapest (PM$_{2.5}$; Salma et al., 2017) during the heating period.

### 3.2.4. Comparison of nucleation particle number concentrations during lockdowns to the reference years

Mean PNCs from factors attributed to nucleation, during reference and lockdown periods, were amongst the largest of the resolved factors, irrespective of environment type (**Figure 5** and **Table 3/Table 4**). The response of these factors to lockdown restrictions were varied.

Among urban background sites, mean PNCs from nucleation factors were higher at HEL_UB and LEJ_UB, and lower at BUD_UB, during lockdown compared to the equivalent periods in the reference years (**Figure 5** and **Table 3**). Changes to mean PNCs from nucleation factors likely reflect variations to photochemical nucleation, as well as reductions to primary aerosol emissions and secondary aerosol formation from vehicle exhaust. The former is likely driven by sulphuric acid, bases such as dimethylamine, and OOMs (Lee et al., 2019). Variations in solar radiation (**Figure S7**) could



also influence NPF via photochemical processes (Shen et al., 2021). The insolation was markedly higher in London during the lockdown period than in previous years, but the other cities show only a small increase. Strong solar radiation favours OH production which through various formation and

420 oxidation processes produces sulphuric acid and other low volatility vapours in the atmosphere (Wang et al., 2023). The latter is linked to the number of vehicles on the road, as well as the associated emissions technologies, which can significantly impact the formation mechanisms and composition of emitted nanocluster aerosol (Rönkkö et al., 2017). Vehicle exhaust also contains significant amounts of nucleation mode particles which directly affect PNCs in urban and suburban areas

(Rönkkö et al., 2017). Importantly, relatively high concentrations of pre–existing particles reflected in the condensation sink may inhibit photochemical nucleation processes by scavenging gas–phase molecules and their clusters (Du et al., 2022). The mixed response of nucleation factors at urban background sites to lockdown restrictions likely represents the interplay between these complex variables.

At roadside sites, mean PNCs from nucleation factors were comparable at LDN_RS, and lower at HEL_RS and LEJ_RS, during lockdown compared to the equivalent periods in the reference years (**Figure 5** and **Table 4**). It is important to acknowledge that instrumentation used to sample particle size distribution data at LDN_RS covered a narrower diameter range compared to instruments used

at other monitoring sites included in this study (**Table 2**). A common size range was selected between sites (10–600 nm) to help streamline the analysis but particle sizing instrumentation at LDN_RS fell short of this lower size cut. This discrepancy will have led to lower PNCs at LDN_RS and likely disproportionately impacted the nucleation factor at this location. However, all particle sizers were equipped with an aerosol dryer to limit relative humidity in the sampled air (**SI section 1.1**). Relative

humidity was controlled to minimise diameter changes due to hygroscopic growth and the resultant particle losses in the dryer were characterised and accounted for in the data analysis as recommended by Wiedensohler et al. (2012). The harmonised approach aided in the comparability between measurements, particularly in the lower size cut. Nevertheless, nucleation factors at roadside locations revealed some wide–ranging reductions to their mean PNCs during lockdown. Moreover, nucleation

factors at monitoring sites in close proximity to one and other, but of different environment type, responded differently to lockdown measures. Again, this points to the importance of traffic and its multifaceted contribution to nucleation mode particles. Road traffic leads to an increased condensation sink which has a negative effect on nucleation, but is a source of organic vapours which can rapidly oxidise forming molecules of low volatility which enhance particle growth rates and surviv-

ability (Brean et al., 2023).



### 3.2.5. Comparison of secondary aerosol particle number concentrations during lockdowns to the reference years

Mean PNCs from factors attributed to secondary aerosols ($O_3$–associated SA and SIA), during the reference and lockdown periods, were the smallest of the resolved factors, irrespective of environment type (**Figure 5** and **Table 3/Table 4**). Though, SIA factors were the largest contributors to PMCs (**Figure 2b**). The response of these factors to the lockdown periods were consistent across all of the monitoring sites included in this study.

Mean PNCs from $O_3$–associated SA and SIA factors were lower at urban background and roadside sites during lockdown compared to the equivalent periods in the reference years (**Figure 5** and **Table 3/Table 4**). The response of SIA factors to lockdown restrictions may reflect reductions in gaseous precursor pollutants (**Figure 4**). Mechanisms of secondary aerosol formation changed under lockdown conditions in Beijing, for example, when $NO_x$ levels substantially declined (Yan et al., 2023). SIA is generated by the transfer of inorganic material from the vapour to the aerosol phase following the chemical processing of emitted gas–phase precursor emissions (McFiggans et al., 2015). Both precursors and particles may be emitted locally or transported long distances from adjacent source regions. Road traffic is typically the largest source of $NO_x$ in an urban area, as well as an under–recognised source of ammonia (Cao et al., 2022). Ammonia reacts with acid pollutants such as oxidation products of $NO_x$ and $SO_2$ to form ammonium nitrate and ammonium sulphate which is essential for the generation of SIA in PM (Duan et al., 2021). These interactions may help to explain why changes to mean PNCs from SIA factors were typically more pronounced at roadside sites.

### 4. CONCLUSION

A multivariate factor analysis technique, PMF, was applied to particle number size distribution data to better understand how PNCs and their sources changed during the respective spring 2020 lockdown periods, compared to the equivalent days of the year in the reference years 2014–2019, depending on data coverage. The analysis involved six monitoring sites (three urban background and three roadside) between four European cities, including Helsinki, Leipzig, Budapest, and London. A number of common factors were resolved between the different sites, including nucleation, road traffic$_{svf}$, road traffic$_{sf}$, diffuse urban, $O_3$–associated SA, and SIA. Despite their commonalities, factors exhibited varying profiles between sites, illustrative of the complex network of aerosol sources and sinks contributing to particle size distributions in urban areas.





The factors attributed to nucleation, road traffic, and diffuse urban were the largest contributors to mean PNCs, during the reference years and the respective lockdown periods. Total mean PNCs were

lower at two of the three urban background sites and at all of the roadside sites during lockdown compared to the reference years. Nucleation factors showed highly variable behaviour. This perhaps demonstrates the important contribution from traffic to nucleation mode particles – either through the direct emission of primary aerosol or via key precursor compounds, such as amines and organic molecules. Road traffic$_{svf}$ factors were also highly variable. This likely reflects the complex interplay

between decreased precursor emissions and a lower condensation/coagulation sink giving variable outcomes. It is also possible that reduced traffic volumes and economic activities were partly counterbalanced by increased domestic emissions. Mean PNCs from road traffic$_{sf}$, on the other hand, were notably lower at roadside locations. The response of diffuse urban factors to lockdown measures were largely consistent and perhaps reflect the more social and cultural aspects of urban air pollutant emis-

sions. Secondary aerosols (O$_3$–associated SA and SIA) exhibited extensive reductions to their mean PNCs during lockdown at all sites. However, SIA remained the largest contributor to PMCs.

The analyses reveal a complex and varied response in the particle size distributions to the curtailment of human mobility during the COVID–19 lockdown periods. The analyses also offer a glimpse into

the future, where the electrification of road transport, together with traffic reduction schemes, may reduce mean PNCs from traffic, as well as potentially shift the relative importance of other sources in urban areas, driving the need for further air quality interventions and policy changes.

Furthermore, as alluded to in this study, the literature encompasses a wide range of named factors,

often characterised by substantial overlap, particularly when it comes to factors related to road traffic. We argue that the named factors introduced in this study describe the reality better than variants present in other works and should be used going forward.

**DATA AND MATERIALS AVAILABILITY**

Data supporting this publication are openly available from the UBIRA eData repository at

https://doi.org/10.25500/edata.bham.00001040

**AUTHOR CONTRIBUTIONS**

Conceptualisation – JB; data curation and/or resources – TP, MV, IS, JN, HM and DvP; formal anal-

ysis – AR and JB; funding acquisition – RH; investigation – AR and JB; methodology – JB; project administration – RH; software – JB and DB; supervision – RH and ZS; visualisation – AR, JB, and



DB; writing (original draft preparation) – AR; writing (review & editing) – JB, DB, ZS, RH, TP, IS, JN, and DvP.

**COMPETING INTERESTS**

At least one of the (co-)authors is a member of the editorial board of Atmospheric Chemistry and Physics.

**ACKNOWLEDGEMENTS**

University of Birmingham acknowledges National Physical Laboratory (NPL) for facilitating access to the London Marylebone Road datasets.

Hungarian Research, Development and Innovation Office (grant K132254).

Dr. Pasi Aalto, University of Helsinki, is acknowledged for his work on maintaining and developing size distribution measurements in Helsinki sites.

Financial support of University of Helsinki through ACTRIS–HY and Research Council of Finland via Atmosphere and Climate Competence Center (ACCC, grant number 337549), RI–URBANS project (Research InfrastructuresServices Reinforcing Air Quality Monitoring Capacities in European Urban & Industrial Areas, European Union's Horizon 2020 research and innovation programme, Green Deal, European Commission, under grant agreement No 101036245), and Urbaani Ilmanlaatu

2.0 via Technology Industries of Finland Centennial Foundation.

Natural Environment Research Council (R8/H12/83/011 and NE/V001523/1).
Katja Moilanen from the City of Helsinki is acknowledged for the traffic count data of Mäkelänkatu.

TROPOS acknowledges technical support of the measurements by René Rabe and Anett Dietze.




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
