# Peer review of "Insights into the sources of ultrafine particle numbers at six European urban sites obtained by investigating COVID–19 lockdowns"

_EGUsphere, 2023_

## Referee Comment (RC1)

**Summary:**

The authors applied comprehensive analysis to study the emission sources, generation mechanisms, and potential sinks of urban particulate matter (PM) through the particle number and mass concentration data from observation sites in four European cities. By comparing the PM concentration differences between the lockdown and reference periods, the authors analyzed the impact of COVID-induced lockdowns on emission intensity and atmospheric physicochemical processes, and consequently, the changes in PM concentrations. The apportionment of the sources/generation mechanisms of PM was quantified by positive matrix factorization (PMF) approach. This study found that nucleation, road traffic, and diffuse urban emissions were the dominant sources to mean PM number concentration, while formation of secondary inorganic aerosol contributes most to the PM's mass concentration. Lockdown was found to have variated impacts on the abovementioned PM sources and mechanisms.

**General suggestions:**

1. This paper studied the impacts of lockdown on PM concentration and their determinant factors. Given the lockdown starting/ending date in each country is varied, it would be better to add a figure to illustrate what principle/index you used here to determine the lockdown time periods for four countries (i.e., the proxy vehicle mobility data or other ancillary index).
2. The daily and weekly cycles of each major factor in Fig 3. represent the typical emission/formation patterns of PM. Does this pattern change during the lockdown period? A comparative figure between lockdown and reference year on different time-scale cycles may help to illustrate the shift of anthropogenic activities and its impact on PM concentration.
3. Do current PMF factors quantify the contributions of transported or aging non-anthropogenic PM from upwind regions? Or any of the additional factors can be added to partially explain the contribution of PM's mid-/long-range transport from non-anthropogenic emissions.
4. The deweathering/detrending technique is needed especially comparing the lockdown effects on pollutant concentration. The authors have stated that such technique has not been applied here because some of the PMF factors are associated with meteorological parameters. It would be recommended to add a similar PMF analysis on the deweathered data to exclude the effects of interannual variations on PNC or PMC. The factors such as road traffic solid fraction is less biased after conducting detrending process.

**Line-by-line suggestions:**

1. Line-107 (Table 1 title): May use the full name of ACTRiS when it first appears.
2. Line-228 & 232 (Figure 3 & 4 title): Briefly explain how the G values used here was computed. What does the magnitude of G values stand for.
3. Line-232 (Figure 4 title): The figure indicates the normalized G values. Briefly introduce how does the normalization was conducted.
4. Figure 2 & 3: Recommend to enlarge the figure and adjust the image layout for a better visualization of the results. The shaded area in Fig. 3 is a bit difficult to distinguish.
5. Figure S7, temperature row: Is the magnitude of temperature in this figure the absolute or relative temperature?

---

## Author Response (AR1)

**Insights into the sources of ultrafine particle numbers at six European urban sites obtained by investigating COVID–19 lockdowns. Response to reviewers.**

Note: Review comments are displayed in plain text, responses to those comments are displayed in blue and sections that have been added to the text are coloured *green (and italicised)* We thank the reviewers for their insightful comments and provide responses below.

**Reviewer: 1**

**Summary:**

The authors applied comprehensive analysis to study the emission sources, generation mechanisms, and potential sinks of urban particulate matter (PM) through the particle number and mass concentration data from observation sites in four European cities. By comparing the PM concentration differences between the lockdown and reference periods, the authors analyzed the impact of COVID-induced lockdowns on emission intensity and atmospheric physicochemical processes, and consequently, the changes in PM concentrations. The apportionment of the sources/generation mechanisms of PM was quantified by positive matrix factorization (PMF) approach. This study found that nucleation, road traffic, and diffuse urban emissions were the dominant sources to mean PM number concentration, while formation of secondary inorganic aerosol contributes most to the PM's mass concentration. Lockdown was found to have variated impacts on the abovementioned PM sources and mechanisms.

**General suggestions:**

1. This paper studied the impacts of lockdown on PM concentration and their determinant factors. Given the lockdown starting/ending date in each country is varied, it would be better to add a figure to illustrate what principle/index you used here to determine the lockdown time periods for four countries (i.e., the proxy vehicle mobility data or other ancillary index).

COVID lockdowns are defined as the periods where mixing in public spaces is prohibited. Lockdowns were phased in their introductions and removals differently in each country, and in most cases were staged. We do not have reliable traffic data at each site, although we agree this would be a good metric. For each country, we chose the start as the point where legal measures were imposed (typically closures of schools), and the end where economic or social activity was reintroduced. No single metric was used, as this would lead to vastly different "lockdown" periods for each country (for example, Budapest kept schools shut longer than Germany). Using periods of different numbers of months would then lead us into problems of differing meteorology. For example, in the UK, the start is defined as when the official government lockdown was introduced, and the end as when non-essential shops opened, spurring movement of people. We include this in the below discussion

*"As the COVID lockdown was different in each country, we chose the start point as where a substantial limitation on human movement was imposed around mid-March, and the end of the period as when some large change was made to human movement in mid-May. Using any single metric*

*would lead to vastly different sampling periods where differences in meteorology would dominate our*
*results, rather than differences in emissions.”*

2. The daily and weekly cycles of each major factor in Fig 3. represent the typical emission/formation
patterns of PM. Does this pattern change during the lockdown period? A comparative figure between
lockdown and reference year on different time-scale cycles may help to illustrate the shift of
anthropogenic activities and its impact on PM concentration.

Great suggestion, thank you. We include this figure in the supplementary as Figure S5 and include the
following discussion in the paper (new text in bold)

*"The response of these factors to lockdown measures were largely consistent, and the diurnal cycle of*
*each source-related factor is similar for both the lockdown and reference years (Figure S5)”*

[Figure]

*Figure S5: Diurnal cycle of G values from the PMF model for each factor for the reference and*
*lockdown periods.*

3. Do current PMF factors quantify the contributions of transported or aging non-anthropogenic PM
from upwind regions? Or any of the additional factors can be added to partially explain the
contribution of PM's mid-/long-range transport from non-anthropogenic emissions.

In an urban area, and in particular at a roadside, the influence of natural PM will be small relative to
the anthropogenic. As there is no chemical speciation in the MPSS data, but simply size and number
data, it is difficult to isolate factors for naturally occurring aerosols. This is easier when analysing
chemical data, in particular when chemical tracers are used (Yin et al., 2015). However, as biogenic
VOC concentrations are small relative to anthropogenic VOC concentrations, secondary organic
aerosol is mostly dominated by AVOCs. Similarly, anthropogenic nitrogen and sulphur emissions will
dominate the inorganic fractions of secondary aerosol, therefore, natural secondary aerosol likely gets
identified in the PMF model as one of our secondary aerosol factors, and makes a small contribution.
Similarly, natural biomass burning would be indistinguishable from anthropogenic biomass burning
that is also aged, but our sampling sites are far from any natural fires and we believe the influence of
aged biomass burning is low. Sea salt makes a small contribution to the PNSD, and usually requires
some dedicated work to pick apart in PNSD PMF, even at coastal sites (Xu et al., 2024), and
therefore, we believe sea salt is a small contributor. We have no way of quantifying what fraction of
our NPF is natural vs biogenic, but model studies would indicate that most particle formation is
influenced by anthropogenic emissions (Gordon et al., 2017), and particle growth is likely driven by
the products of AVOC oxidation (Li et al., 2022). We acknowledge these uncertainties in the
following lines when discussing secondary aerosol:

*"In an urban environment it is reasonable to presume that most SA precursors are anthropogenic, but*
*an influence of natural SA precursors will also contribute some fraction of total SA."*

When discussing sea salt and dust:

*"We do not expect a substantial contribution from dust and sea salt in the particle number size*
*distribution at diameters <600 nm."*

When discussing NPF:

*"NPF in these urban areas is likely driven by sulphuric acid, bases such as dimethylamine, and*
*OOMs, likely from AVOC oxidation (Lee et al., 2019), meaning our NPF is mostly driven by*
*anthropogenic emissions"*

And when discussing our diffuse urban factors:

*"A contribution of aged natural aerosol may also be present in this factor."*

4. The deweathering/detrending technique is needed especially comparing the lockdown effects on
pollutant concentration. The authors have stated that such technique has not been applied here because
some of the PMF factors are associated with meteorological parameters. It would be recommended to
add a similar PMF analysis on the deweathered data to exclude the effects of interannual variations on
PNC or PMC. The factors such as road traffic solid fraction is less biased after conducting detrending
process.

We agree that deweathering leads to better insights when applied to pollutant data such as NO$_x$, CO,
etc. (Shi et al., 2021). Deweathering mostly accounts for the dispersion of pollutants due to dilution
(heightening of the boundary layer, dilution due to wind speeds etc.). In the case of the PNSD
however, the effects of meteorology are more complex. Many PM components are semivolatile, and
increases in temperature can lead to their evaporation. Similarly, high temperatures inhibit rapid NPF
as they increase the evaporation rate of small clusters. However, high temperatures are often
coincident with intense solar radiation, which will accelerate the formation of sulphuric acid and other
NPF precursors. Emissions of SOA precursors from processes such as evaporation of VOCs from
traffic vehicles is also temperature dependent (Cliff et al., 2023). We therefore argue the dynamics of
the PNSD are too complex to try and apply deweathering to. For this reason, we included the
following text in the manuscript, which we have updated with extra detail.

*"In examining temporal changes affecting pollutant concentrations, it is now common practice to*
*remove the influences of changes in weather variables which affect primary pollutant concentrations*
*(Vu et al., 2019; Shi et al., 2021) primarily by accounting for dilution effects. Such a treatment was*
*not applied in this study as weather variables affect both photochemical nucleation, with elevated*
*temperatures accelerating the evaporation rate of clusters, and intense solar radiation accelerating*
*the generation of NPF precursors (Lee et al., 2019; Bousiotis et al., 2021), and the semi-volatile*
*fraction of both traffic and secondary aerosols through evaporation (Charron et al., 2003). Similarly,*
*emissions of organic aerosol precursors are temperature-dependent (Lee et al., 2019). The dynamics*
*of the number size distribution are therefore more complex than can be accounted for by de-*
*weathering methods, and such results could not be interpreted with confidence. Rather, the trends in*
*meteorological variables are presented in **Figure S8**, and commented upon in the text, where*
*considered relevant."*

As well as all associated discussion surrounding Figure S8

**Line-by-line suggestions:**

1. Line-107 (Table 1 title): May use the full name of ACTRiS when it first appears.

Great suggestion. This has been done.

2. Line-228 & 232 (Figure 3 & 4 title): Briefly explain how the G values used here was computed.
What does the magnitude of G values stand for.

We now finish the titles of Figure 3 and 4 with the following:

*"The G value is the time-series component of the PMF solution (see supplement)."*

3. Line-232 (Figure 4 title): The figure indicates the normalized G values. Briefly introduce how does
the normalization was conducted.

The title of Figure 4 now contains the following:

*"The normalisation is performed so each variable has a maximum value of 1 for easy comparison."*

4. Figure 2 & 3: Recommend to enlarge the figure and adjust the image layout for a better
visualization of the results. The shaded area in Fig. 3 is a bit difficult to distinguish.

We have made two edits to the figures. 1) we have made the shaded region in Fig. 3 darker for
legibility; 2) we have printed all images as 600 dpi images for clarity.

5. Figure S7, temperature row: Is the magnitude of temperature in this figure the absolute or relative
temperature

Thanks for pointing out the error – the units are in fact in degrees centigrade, not Kelvin. They are
absolute, and we have now amended the figure.

**Reviewer: 2**

The authors applied PMF to elucidate how particle number concentrations evolve during spring 2020 lockdown period relative to the reference years of 2014-2019. While the topic is overall interesting, I feel the novelty is very weak in the current version, and a substantial revision is needed to satisfy the standard of the journal.

1. This study heavily relies on PMF method. Is there any uncertainty? The authors seem to consider the source apportionment based on PMF as a fact, while I think there might be large uncertainties associated with this method.

The main uncertainty with PMF, especially as applied to particle number size distribution (PNSD) data is the attribution of factors. As we do not have chemical tracer data (Yin et al., 2015), the assignment of factors is based upon the shape of the size distribution and the diurnal and annual cycles of each factor. There is a little ambiguity here, as sometimes several sources blend into one. This is discussed in the paper extensively in section 3.1.1., where new particles from NPF and traffic blend together into a "nucleation" factor, section 3.1.2 where non-traffic combustion emissions contribute to the "traffic$_{(svf)}$ factor, section 3.1.4., where "diffuse urban" is a combination of aged emissions, and section 3.1.5 plus 3.1.6 where the origin of the secondary aerosol is unknown. This is discussed extensively in section 3.1. Further, there is the problem of rotational ambiguity, which is currently not discussed in the manuscript. We now include the following in the discussion of PMF methods:

*"Rotational ambiguity arises in PMF as the decomposition of the data into G and F is not unique; there can be multiple pairs of G and F matrices that satisfy the model while producing the same quality of fit. However, PMF inherently limits this ambiguity through the imposition of non-negativity constraints on both G and F. This restriction significantly reduces the degrees of freedom for rotation, as it allows only transformations that maintain non-negative entries, thereby ensuring that all extracted factors and their contributions remain physically interpretable. Despite this, different rotations that meet the non-negativity conditions can still produce valid solutions, potentially leading to different interpretations of the source contributions and profiles. These can be investigating using FPEAK (Paatero and Tapper, 1994), however, in this work the initial solutions were deemed physically meaningful, and changing FPEAK did not improve our results."*

*Paatero, P. and Tapper, U.: Positive matrix factorization: A non-negative factor model with optimal utilization of error estimates of data values, Environmetrics, 5(2), 111–126, doi:10.1002/env.3170050203, 1994.*

2. The overall comparison between lockdown and reference period seems to be very general and not surprising. It looks like many statements has been known already.

We argue that the new findings in this manuscript satisfy the criteria for novelty of ACP. We provide the first PMF study of particle number size distributions in Europe during COVID lockdowns, and we perform this across six sites. The smaller-than-expected reduction in traffic during COVID lockdowns points to a complex picture of pollutant reduction under traffic reduction. The increase in traffic emissions and PNCs at the LEJ_UB site points to a possible misattribution of "traffic" factors in the literature, as stationary combustion sources may be more important than previously thought. This is only evident when looking at the data during the COVID lockdown periods, and we therefore advance the field of PNSD PMF by advocating new naming conventions. The mixed response of the nucleation factor points to the complexity of the process: primary emissions both accelerate NPF through the emissions of precursors, and slow NPF through the emission of particles with a high surface area. The same applies to primary traffic nucleation particles, where their lifetimes increase as total PM surface area declines. The consistent decline of secondary aerosols is also novel, and can likely be attributed to the reduction in precursor emissions. Further to this, this study extends previous analyses (Rivas et al., 2020) to new sites.

3. In many sections, there are only a couple of sentences, briefly describing the difference between lockdown and reference period. These short paragraphs are not acceptable in my opinion, and they do not raise any scientific finding in depth.

We provide data in Tables 3 and 4, Figures 5 and 6, and in an updated Figure S5 below. We do not provide a detailed discussion of the time series of each factor, as the diurnal cycle is the same for each factor in the reference and lockdown years. We highlight this fact in the following sentence:

*"The response of these factors to lockdown measures were largely consistent, **and the diurnal cycle of each source-related factor is similar for both the lockdown and reference years (Figure S5)**"*

In each section from 3.2.1. to 3.2.5. we provide 2-3 paragraphs of discussion of each factor. This contains both discussion of the changes in magnitude, and speculation about the possible causes. As this review does not contain any particular suggestions for what we should be describing, we have opted to add more detail about the magnitude of change (this is included in bold in the below sections)

*"Among urban background sites, mean PNCs from road traffic$_{svf}$ factors were higher at LEJ_UB, comparable at HEL_UB, and lower at BUD_UB during lockdown compared to the equivalent periods in the reference years, **with a mean change of +3.7% across the three sites (Error! Reference source not found. and Error! Reference source not found.).**"*

*"At roadside sites, mean PNCs from road traffic$_{svf}$ factors were comparable at HEL_RS, and lower at LEJ_RS and LDN_RS during lockdown compared to the equivalent periods in the reference years, **with a mean decrease of -21% (Error! Reference source not found. and Error! Reference source not found.).**"*

*"Among urban background sites, mean PNCs from diffuse urban factors were higher at LEJ_UB, and lower at HEL_UB and BUD_UB, during lockdown compared to the equivalent periods in the reference years, **with a mean decrease of -11% (Error! Reference source not found. and Error! Reference source not found.).** Mean PNCs from the factor were also lower at LEJ_RS during lockdown **by -16% (Error! Reference source not found. and Error! Reference source not found.).**"*

*"Among urban background sites, mean PNCs from nucleation factors were higher at HEL_UB and LEJ_UB, and lower at BUD_UB, during lockdown compared to the equivalent periods in the reference years **with a mean increase of 13%"***

*"At roadside sites, mean PNCs from nucleation factors were comparable at LDN_RS, and lower at HEL_RS and LEJ_RS, during lockdown compared to the equivalent periods in the reference years, **with a mean decrease of -43%"***

*"Mean PNCs from O$_3$–associated SA and SIA factors were lower at urban background and roadside sites during lockdown compared to the equivalent periods in the reference years, **with mean***

*decreases of -42% in SIA at the urban background sites, and -42% and -60% for O₃-associated SA*
*and SIA at the roadside"*

[Figure]

**Figure S5:** *Diurnal cycle of G values from the PMF model for each factor for the reference and lockdown periods.*

4. There are no mechanisms at all besides of the general descriptions. I feel the authors should at least try to add some mechanism discussion, not just describe things based on PMF, and cite a few papers without any deeper understanding of the changes in PNCs.

In our view, we have included extensive mechanistic discussion, and provide sufficient reasoned speculation to explain what we observe in the data. We do not provide detailed chemical mechanisms, but we have not measured any chemical data, and we cite appropriate literature where there is plentiful mechanistic discussion. As this review contains no specific suggestions, we have updated with some extra discussion where we deem it necessary. New text below in bold. In our discussion of traffic$_{svf}$:

*"Lockdown restrictions had immediate and varied impacts on energy use, with increased residential demand due to people being confined to their homes.* ***This points to a systematic misattribution of this factor throughout the literature, where this traffic factor contains some contribution from domestic heating****."*

In our discussion of nucleation:

***NPF is inhibited by high particle surface area, which was lower during lockdowns at all sites except LEJ_UB (Figure 6) (Du et al., 2022). Primary and delayed nucleation particles*** *are linked to the number of vehicles on the road, as well as the associated emissions technologies, which can significantly impact the formation mechanisms and composition of emitted nanocluster aerosol (Rönkkö et al., 2017). . The mixed response of nucleation factors at urban background sites to lockdown restrictions likely represents the interplay between these complex variables.*

***Across all urban background sites, the largest increase to nucleation was at LEJ_UB, 92% increase, where the greatest increase to traffic factors was also seen (51% increase to traffic***$_{svf}$***), while CS also increased (Figure 6). The largest decrease in nucleation was at BUD_UB, 45% decrease, where the greaste reduction in traffic factors was also seen (-36% decrease to traffic***$_{svf}$***), while CS decreased. Primary nanocluster aerosol will have a lifetime on the order of tens of minutes, and will need to grow to 10 nm and also undergo dilution while being transported from the roadside to the urban background measurement stations. Primary and delayed primary particle emissions will be less important here than at the roadside, and we infer some substantial role of primary gaseous traffic emissions in accelerating NPF (Brean et al., 2023).***

***"The large decrease to nucleation at the roadside can be attributed either to a decrease in primary nanocluster aerosol emissions, or to a reduction in NPF precursor emissions (Brean et al., 2023)."***

In our discussion of secondary aerosols:

*"…These interactions may help to explain why changes to mean PNCs from SIA factors were typically more pronounced at roadside sites.* ***Similarly, if O***$_3$***-associated SA is generated through ozonolysis of VOCs, VOC concentrations will have declined substantially during lockdown periods, although, O***$_3$ ***concentrations increased during lockdown periods (Shi et al., 2021). This decline to secondary aerosol is large, consistent across all sites, and will result in a substantial reduction to PM mass."***

5. Figure quality is very poor. Most figures are not easy to read and should be redrawn to be legible.

Thank you for highlighting this. We have presented all figures as 600 dpi .jpg images and these will also be uploaded as vector images with final submission.

**Minor comments**

1. Line 417: The insolation was markedly higher in London during the lockdown period than in previous years, but the other cities show only a small increase. Any reasons why London has much higher downward solar radiation over this period? Why not in other cities?

Yes. It is similarly high in HEL_RS. The lockdown period in southern England was characterised
by unusually sunny weather. We can't find any publications discussing this.

2. O3–associated SA and SIA: What is ozone associated SIA? It is very vague, and the readers
would not know what it is. Is it secondary organic or inorganic aerosols? Is SIA also O3
associated?

$O_3$-associated SA has been called $O_3$-rich SA in the previous literature (e.g., Rivas et al., 2020),
which we re-name, as the latter implies the aerosol itself is rich in ozone. We have no way of
knowing whether it is an organic or inorganic aerosol, as we have no measurements of aerosol
chemical composition. This is a commonly found PMF factor in PNSD PMF, where secondary
aerosols are highly associated with high $O_3$ concentrations. Other papers have inferred that this
may be the condensation of organic ozonolysis products, but we do not want to infer too much
chemistry when we have no chemical data. This is discussed in section 3.1.5. As shown in Figure
4, SIA is not associated with $O_3$ except when there exists no $O_3$-associated SA.

**References**

Cliff, S. J., Lewis, A. C., Shaw, M. D., Lee, J. D., Flynn, M., Andrews, S. J., Hopkins, J. R., Purvis,
R. M., and Yeoman, A. M.: Unreported VOC Emissions from Road Transport Including from Electric
Vehicles, Environmental Science & Technology, 57, 8026-8034, 10.1021/acs.est.3c00845, 2023.
Gordon, H., Kirkby, J., Baltensperger, U., Bianchi, F., Breitenlechner, M., Curtius, J., Dias, A.,
Dommen, J., Donahue, N. M., Dunne, E. M., Duplissy, J., Ehrhart, S., Flagan, R. C., Frege, C., Fuchs,
C., Hansel, A., Hoyle, C. R., Kulmala, M., Kürten, A., Lehtipalo, K., Makhmutov, V., Molteni, U.,
Rissanen, M. P., Stozkhov, Y., Tröstl, J., Tsagkogeorgas, G., Wagner, R., Williamson, C., Wimmer,
D., Winkler, P. M., Yan, C., and Carslaw, K. S.: Causes and importance of new particle formation in
the present-day and preindustrial atmospheres, Journal of Geophysical Research: Atmospheres, 122,
8739-8760, 10.1002/2017jd026844, 2017.
Li, X., Li, Y., Cai, R., Yan, C., Qiao, X., Guo, Y., Deng, C., Yin, R., Chen, Y., Li, Y., Yao, L.,
Sarnela, N., Zhang, Y., Petäjä, T., Bianchi, F., Liu, Y., Kulmala, M., Hao, J., Smith, J. N., and Jiang,
J.: Insufficient Condensable Organic Vapors Lead to Slow Growth of New Particles in an Urban
Environment, Environmental Science & Technology, 56, 9936-9946, 10.1021/acs.est.2c01566, 2022.
Rivas, I., Beddows, D. C. S., Amato, F., Green, D. C., Järvi, L., Hueglin, C., Reche, C., Timonen, H.,
Fuller, G. W., Niemi, J. V., Pérez, N., Aurela, M., Hopke, P. K., Alastuey, A., Kulmala, M., Harrison,
R. M., Querol, X., and Kelly, F. J.: Source apportionment of particle number size distribution in urban
background and traffic stations in four European cities, Environment International, 135, 105345,
https://doi.org/10.1016/j.envint.2019.105345, 2020.
Shi, Z., Song, C., Liu, B., Lu, G., Xu, J., Van Vu, T., Elliott, R. J. R., Li, W., Bloss, W. J., and
Harrison, R. M.: Abrupt but smaller than expected changes in surface air quality attributable to
COVID-19 lockdowns, Science Advances, 7, eabd6696, 10.1126/sciadv.abd6696, 2021.
Xu, W., Zhong, H., Lin, C., Huang, R.-J., Ovadnevaite, J., Ceburnis, D., and O'Dowd, C.:
Identification of Sub-micrometer Ambient Sea Salt Number Size Distribution by Positive Matrix
Factorization, ACS ES&T Air, 10.1021/acsestair.3c00092, 2024.
Yin, J., Cumberland, S. A., Harrison, R. M., Allan, J., Young, D. E., Williams, P. I., and Coe, H.:
Receptor modelling of fine particles in southern England using CMB including comparison with
AMS-PMF factors, Atmos. Chem. Phys., 15, 2139-2158, 10.5194/acp-15-2139-2015, 2015.